# Recent Developments in the Photochemical Synthesis of Functionalized Imidazopyridines

**DOI:** 10.3390/molecules27113461

**Published:** 2022-05-27

**Authors:** Christine Tran, Abdallah Hamze

**Affiliations:** Department of Medicinal Chemistry, CNRS, BioCIS, Université Paris-Saclay, 92290 Châtenay-Malabry, France

**Keywords:** imidazopyridines, visible light, photochemistry

## Abstract

Imidazopyridines constitute one of the most important scaffolds in medicinal chemistry, as their skeleton could be found in a myriad of biologically active molecules. Although numerous strategies were elaborated for imidazopyridine preparation in the 2010s, novel eco-compatible synthetic approaches have emerged, conscious of climate change concerns. In this framework, photochemical methods have been promoted to conceive this heterocyclic motif over the last decade. This review covers the recently published works on synthesizing highly functionalized imidazopyridines by light induction.

## 1. Introduction

Imidazopyridines have had a long-lasting interest in organic and medicinal chemistry [1,2,3]. These heterocyclic scaffolds have broadly been found in many pharmaceuticals with various biological activities. For example, saripidem or alpidem are hypnotic drugs that are chemically distinct from benzodiazepines but bind at the same site on the receptor. Zolpidem, more commonly known as Ambien^®^, is employed as a medication in the therapy for insomnia. Zolimidine, a gastroprotective agent, also exhibits an imidazopyridine motif in its structure. More recently, researchers considered further biological applications for these molecules, namely involving their antibacterial [4], analgesic, anti-inflammatory [5], antiviral [6], and antitumor [7] properties (Figure 1). Besides, imidazopyridines could also show significant fluorescent properties, which could be utilized to construct polymer films [8].

Given their importance in applied sciences, tremendous efforts have been conducted since the last century to prepare these *N*-heterocycles [1,9]. In 1925, Tschitschibabin and Kirsanow were the first to synthesize imidazopyridines by heating 2,3-aminopyridine derivatives with acetic anhydride [10]. This skeleton was ignored for a long time in organic chemistry due to the underdevelopment of efficient methodologies for accessing highly functionalized molecules. In recent decades, the upturn of organometallic chemistry has emulated numerous synthetic strategies, particularly C–H functionalizations [11,12,13,14,15], condensations [16], or multicomponent reactions [17]. With awareness of the current environmental issues and the decrease in non-renewable sources, the development of straightforward and mild procedures using eco-friendly conditions is highly desirable. Thus, in the 2010s, photochemical methods applied to imidazopyridine platforms have expanded intensively [18,19,20,21]. Photochemistry offers many advantages over conventional heating, with the use of light as a sustainable energy source and less toxic organic reagents or catalysts [22]. In this context, visible-light-induced approaches for synthesizing and functionalizing imidazopyridines have flourished in the last decade [23,24,25]. Various photocatalytic reactions have been implemented with metallic [26,27] or metal-free [28,29,30,31,32] catalysts.

As an overview of the recent updates in imidazopyridine chemistry, the review includes the literature survey up to April 2022. We have detailed original transformations of the imidazopyridine’s building blocks via C–H functionalization or multicomponent, or tandem reactions.

## 2. C–H Functionalization

In 2015, the Hajra group developed the first metal-free C–H thiocyanation of imidazo[1,2-*a*]pyridines, with eosin Y as the photocatalyst and air as a green oxidant, in acetonitrile (ACN) and under a blue LED (light-emitting diode). A wide range of substituted 3-(thiocyanato)imidazo[1,2-*a*]pyridines was afforded, with high yields. The methodology was also extended to selenocyanation and trifluoromethylthiolation reactions (Figure 1) [33]. By the control experiments, the C–H functionalization could occur via the visible-light photoactivation of eosin Y, forming the thiocyanate radical. This latter intermediate could react with the imidazo[1,2-*a*]pyridine to deliver the desired compound after an oxidation and deprotonation sequence (Figure 1).

This first incursion in the C–H functionalization of imidazopyridines has paved the way for diverse transformations applying this approach.

### 2.1. Formation of C–C Bonds

The C–C bond construction in a radical pathway represents one of the significant tools in organic chemistry. In this frame, numerous methodologies have risen over the last decade for the C–C bond’s elaboration at position C_3_ in imidazopyridines.

#### 2.1.1. Fluoroalkylation of Imidazopyridines

Fluorine constitutes a highly privileged bioisostere of the hydrogen atom due to its metabolic stability and lipophilicity [34]. These interesting properties have promoted the incorporation of fluorinated motifs into organic substrates, potential biologically active compounds, or drug candidates [35].

In 2020, Cui et al. detailed the visible-light-mediated metal-free C_3_–H trifluoromethylation of imidazo[1,2-*a*]pyridines, using an acridinium derivative as the photoredox catalyst and Langlois’ reagent (CF_3_SO_2_Na) as the fluorinating agent (Figure 2) in dichloroethane (DCE) [36]. This straightforward procedure is very compatible with electron-rich or electron-poor substituted substrates (up to 84% yield). By TEMPO ((2,2,6,6-Tetramethylpiperidin-1-yl)oxyl) radical capture, the authors proved the involvement of a fluoroalkyl radical intermediate engendered by single electron transfer (SET) via the acridinium photocatalyst.

Another synthetic method consists of a trifluoromethylation with Langlois’ reagent, 4,4′-dimethoxybenzophenone as the photocatalyst, and HFIP (hexafluoroisopromanol) as an additive in dry ACN. In this manner, Lefebvre, Hoffmann, and Rueping developed a C_3_-substituted imidazo[1,2-*a*]pyridine scaffold with a 42% yield (Figure 3) [37].

The Zhang team proposed a regioselective C–H trifluoromethylation in position C_3_ of imidazo[1,2-*a*]pyridines. The investigation of the reaction conditions showed that anthraquinone-2-carboxylic acid (AQN) was the best photocatalyst, employed simultaneously with Langlois’ reagent, trifluoroacetic acid (TFA), and potassium carbonate in DMSO (dimethyl sulfoxide) (Figure 4). This method allowed for access to 21 trifluoromethylated imidazo[1,2-*a*]pyridine derivatives, with moderate-to-good yields. The process’s applicability was validated by the C_3_-trifluoromethylation of Zolimidine, an antiulcer drug, with 55% yield. Zhang and co-workers demonstrated the radical reaction process through mechanistic studies with radical-trapping experiments [38].

Deng and co-workers conceptualized an efficient process for the regioselective C_3_-trifluoromethylation and perfluoroalkylation of imidazo[1,2-*a*]pyridines. By visible-light photoactivation, a broad array of functionalized imidazo[1,2-*a*]pyridines were prepared, with satisfactory results. The main advantage of this method relies on the use of only an organic base (DBU: 1,8-Diazabicyclo[5.4.0]undec-7-ene) with the fluorinating agent in ACN or *N*-methyl-2-pyrrolidone (NMP). Light on/off experiments and radical trapping reactions suggested that an electron-donor–acceptor (EDA) complex could be formed between DBU with trifluoromethyl (or perfluoroalkyl) iodide. The blue-LED irradiation of the EDA complex led to the generation of CF_3_^•^ radicals, which could react with the imidazo[1,2-*a*]pyridine substrate, producing the corresponding radical intermediate. This latter compound could undergo an oxidation-deprotonation sequence (Path A, Figure 5) or a hydrogen abstraction by iodine radicals, delivered from the EDA complex and iodide (Path B, Figure 5) [39].

The same year, Wu and his colleagues developed a similar idea, using DMSO as a solvent instead of NMP for the C_3_-perfluoroalkylation of imidazo[1,2-*a*]pyridines. This modified approach contributed to the synthesis of 27 C_3_-fluorinated imidazo[1,2-*a*]pyridines (Figure 6). A good tolerance is observed for both the electron-withdrawing and electron-donating groups (21 to 96% yield) [40].

The C_3_-trifluoroethylation of imidazo[1,2-*a*]pyridines by Xu and Fu was carried out with *fac*-[Ir(ppy)_3_] (ppy: 2-phenylpyridinato), 1,1,1-trifluoro-2-iodoethane, and K_2_CO_3_ in DMSO [41]. This visible-light-promoted reaction resulted in the preparation of a broad range of C_3_-fluorinated imidazopyridines, exhibiting electron-poor or -rich substituents (Figure 7). Inhibition of this transformation was performed with TEMPO as a radical scavenger; the expected compound was not detected, implying a radical path. The mechanism of this functionalized C–H could thus be rationalized: the oxidation of the excited photocatalyst by CF_3_CH_2_I could lead to the CF_3_CH_2_^•^ radical species. Addition of the latter radical could be accomplished on the imidazo[1,2-*a*]pyridine motif. Oxidation and base-mediated deprotonation could induce the formation of the desired product.

Huang and Zhu went further with the C_3_-perfluoroalkylation of imidazo[1,2-*a*]pyridines with TMEDA (tetramethylethylenediamine) as a radical initiator and K_3_PO_4_ as the base [42]. The transformation displayed a good tolerance with diversely substituted imidazo[1,2-*a*]pyridines (74 to 92% yield). Functional groups in the *meta*- or *para*- position provided the wanted compounds with better results than the imidazo[1,2-*a*]pyridines featuring substituents in the *ortho* position. Under modified conditions, the procedure was also attempted for a C_3_-difluoroacetylation, giving the expected *N*-heterocycle with 61% yield (Figure 8). Mechanistic control experiments with radical scavengers (TEMPO, 1,1-diphenylethene, and hydroquinone) jeopardized the reaction since the expected C_3_-perfluoroalkylated imidazo[1,2-*a*]pyridine was obtained in low yields. With TEMPO, 2,2,6,6-tetramethyl-1-(perfluorobutoxy)piperidine was identified by GC-MS (gas chromatography–mass spectrometry) analysis, confirming the radical character of the process.

To enlarge the diversity of fluorinated imidazopyridines, the Fu team conceived access to (phenylsulfonyl) difluoromethylated structures in the presence of PhSO_2_CF_2_I, K_2_CO_3,_ and *fac*-[Ir(ppy)_3_] [43]. The adopted protocol allowed for the preparation of 15 C_3_-functionalized imidazo[1,2-*a*]pyridines with good-to-high yields (Figure 9).

The same group established a related approach for introducing a difluoroacetyl motif in the C_3_ position of the imidazo[1,2-*a*]pyridine skeleton with BrCF_2_CO_2_Et. Substrates exhibiting electron-donating groups led to the desired products in higher yields than the electron-withdrawing ones (Figure 10) [44].

Xu and co-workers reported the C–H difluoroalkylation of imidazo[1,2-*a*]pyridines mediated by visible light. The protocol requires the use of bromodifluoroaryl ketones as a co-substrate, TMEDA as the organic base in acetonitrile, and a 33 W compact fluorescent light (CFL). These mild and straightforward conditions yielded a wide range of imidazo[1,2-*a*]pyridines displaying various functional groups (Figure 11) [45].

Difluoromethylenephosphonation of imidazo[1,2-*a*]pyridine, realized by the Hajra team, provides functionalized *N*-heterocycles by employing rose bengal (RB) as a photocatalyst, bis(pinacolato)diboron as an additive, and NaHCO_3_ as the base [46]. The exploration of the substrate’s scope revealed that highly substituted imidazo[1,2-*a*]pyridines could be synthesized through this method. The expected products were not observed by attempting the standard reaction with different radical inhibitors (TEMPO, BHT: Butylated hydroxytoluene, *para*-benzoquinone, and 1,1-diphenylethylene), confirming the radical process. Without bis(pinacolato)diboron (B_2_pin_2_), the reaction did not proceed, indicating the crucial role of this additive. With all these findings and cyclic voltammetry measurements, the authors proposed the activation of imidazo[1,2-*a*]pyridine by bis(pinacolato)diboron, generating a cationic intermediate. This intermediate could then undergo the addition of CF_2_PO(OEt)_2_ radicals (formed by RB* oxidation). Hydrogen abstraction by NaHCO_3_ could deliver the difluoromethylenephosphonated imidazo[1,2-*a*]pyridine (Figure 12).

In summary, fluoroalkylation of imidazopyridines could be reached in several reaction conditions, with moderate-to-good yields (up to 96% yield). All the approaches described here required polar aprotic solvents (mainly ACN, DMSO) and organic bases or acids under an inert atmosphere. The photocatalysts employed were organophotocatalysts or *fac*-[Ir(ppy)_3_]. These strategies allowed access to diversely fluorinated compounds.

#### 2.1.2. Alkylation of Imidazopyridines

Alongside the fluoroalkylation of imidazopyridines, the introduction of various moieties by alkylation reactions has also arisen during the last five years. In 2017, inspired by the above-mentioned C_3_-trifluoroethylation of imidazo[1,2-*a*]pyridines by Xu, Fu, and coworkers [41], Liu and Sun developed the C_3_-cyanomethylation of imidazo[1,2-*a*]pyridines using an analogous photocatalytic system [47]. With the inexpensive bromoacetonitrile as a cyanomethyl source, the group efficiently developed a large array of substituted imidazopyridines (up to 96% yield). It should be outlined that a significant yield enhancement was noted for some substrates by employing iodoacetonitrile rather than bromoacetonitrile. This robust method was also applied to the synthesis of Zolpidem and Alpidem, drugs used in anxiety treatment. Once the cyanomethylated imidazo[1,2-*a*]pyridines were isolated, they were converted into the corresponding ethyl esters. These intermediates were then hydrolyzed with KOH and amidified in dichloromethane (DCM) following standard procedures [48], to afford the expected biologically active compounds (Figure 13).

Aminoalkylation has also drawn attention in the topic of imidazopyridines’ functionalization. In 2018, Hajra and co-workers disclosed the metal-free coupling between tertiary amines and imidazo[1,2-*a*]pyridines [49]. With rose bengal as the organocatalyst under aerobic conditions, they combined *N*-phenyltetrahydroisoquinoline with imidazopyridines in a regioselective manner (Figure 14). A broad range of highly substituted imidazopyridines was thus produced. Good to excellent yields were obtained with electron-donating or -withdrawing groups. The approach was also extended to *N*,*N*-dimethylaniline derivatives with success. Control experiments implemented the elucidation of the mechanism with radical scavengers (TEMPO, BHT) and a singlet oxygen quencher (DABCO: 1,4-diazabicyclo[2.2.2]octane). The suggested pathway could pass through an energy transfer between the excited state of the photocatalyst (RB*) and the ground-state oxygen (^3^O_2_). The generated singlet oxygen could undergo an SET for the tertiary amine to deliver the amine radical cation. By hydrogen capture, an iminium is then formed. This latter molecule could be implied in an electrophilic addition with the imidazopyridine. A final proton abstraction could then give the target compound.

More recently, Yu et al. conceptualized a sustainable procedure for the aminomethylation of imidazo[1,2-*a*]pyridines by using *N*-arylglycines as the amine sources and an original metallated photocatalyst (CsPbBr_3_) [50]. The principal advantage of this procedure is the possible re-utilization of the perovskite catalyst for at least five times and with excellent yields (more than 88%). As the reaction is in a heterogeneous system, the recovery of CsPbBr_3_ was facilitated by simple centrifugation. Good compatibility was remarked for substrates featuring donor (Me, OMe, NH_2_) or acceptor (F, Cl, Br, CN, CF_3_, CO_2_Me) substituents. It should be emphasized that aminomethylation is applicable for a gram-scale synthesis with sunlight irradiation. The inhibition of the transformation by radical scavengers suggested a radical reaction mechanism. CsPbBr_3_ could release an electron (e−) and a hole (h+) by absorbing a photon. A SET could then be realized from *N*-arylglycine to the hole, leading to the corresponding radical. This intermediate could then be added to the imidazopyridine scaffold. The oxidation by O_2_ provided the heterocyclic cation, which could evolve to the final product by deprotonation (Figure 15).

The same team went one step further by improving the protocol in a greener way. In 2021, Lv and Yu established an eco-compatible carbon nitride nanosheet (NM-g-C_3_N_4_), which could catalyze under blue-LED irradiation the aminomethylation of imidazopyridines [51]. To fulfill the criteria of green chemistry, dimethyl carbonate was employed this time as the reaction solvent. Again, a set of aminomethylated imidazo[1,2-*a*]pyridines displaying diverse functional groups (18 examples) was elaborated smoothly (Figure 16). As previously, the NM-g-C_3_N_4_ photocatalyst could be reused after the reaction workup by centrifugation. The recycling experiments showed that the photocatalyst’s efficiency is maintained after seven transformation cycles.

Zhu and Le monitored the C–H aminomethylation reaction with *N*-arylglycine derivatives in an analogous eco-compatible way [52]. The reaction occurred efficiently under photocatalyst-free conditions (Figure 17). A wide range of functionalized imidazopyridines was provided with good results (40 to 95% yields). The group unraveled the aminomethylation path with various control experiments, namely, reaction under a nitrogen atmosphere or in an open-air flask, or radical trapping with TEMPO. By blue-LED irradiation, a singlet oxygen could be formed and interact with the *N*-arylglycine substrate to generate a radical cation. This latter intermediate could evolve in an alkyl radical by proton transfer and decarboxylation. Subsequently, the amino radical could undergo a proton transfer, leading to the corresponding imine. The final electrophilic addition of the imine to the imidazopyridine motif allows access to the target product.

The C–H alkylation of imidazo[1,2-*a*]pyridines could be performed with *N*-hydroxyphthalimide esters as alkylating reagents. Jin and his colleagues conceived this original strategy for the C–H functionalization of the aryl part of the imidazopyridine platform [53]. The organic photoredox catalysis implied eosin Y as the photocatalyst and TfOH (triflic acid) as the additive. The reaction was well-tolerated with a wide array of imidazopyridine substrates (up to 86% yield). By checking the *N*-hydroxyphthalimide esters’ scope, satisfactory results were afforded for the primary, secondary, and tertiary alkyl groups. The alkylation pathway was unraveled with radical trapping experiments: an adduct with BHT was identified with HRMS (high-resolution mass spectrometry) analysis, validating the radical mechanism. An SET could occur from the excited state of eosin Y to the protonated *N*-hydroxyphthalimide ester. The formed radical species could be decomposed into an alkyl radical, which could be introduced into the imidazopyrine’s nucleus. The oxidation of the imidazopyridine radical by an SET with eosin Y^•+^ produced the corresponding cation. Finally, the expected compound is obtained via deprotonation (Figure 18).

The Hajra group deepened the concept of C–H alkylation by exploring the three-component carbosilylation of alkenes in the imidazopyridine scaffold [54]. The combination of a metal catalyst (FeCl_2_) and blue-LED photocatalysis enabled the C–C and C–Si bond formation. The reaction involving an imidazopyridine substrate, a styrene derivative, and (TMS)_3_SiH gave a wide array of silylated imidazo[1,2-*a*]pyridines (26 compounds) in 45 to 88% yields (Figure 19). After the scope study, the authors examined the transformation pathway with radical scavengers. The reaction did not occur in the presence of TEMPO, BHT, or benzoquinone, reflecting a radical mechanism. The same result was observed without a photocatalyst or light source. Considering these control experiments, the proposed path could proceed via an SET between the iron(II) catalyst and the excited state of eosin Y. The radical anion eosin Y^•−^ could then realize an SET with the di-*tert*-butyl peroxide, affording the radical *t*BuO^•^. The formed silyl radical will be added to the styrene by hydrogen abstraction. An SET could then be accomplished from the generated radical styrene to iron(III). An electrophilic addition could be reached with imidazopyridine, allowing access to the desired product.

Summarily, the alkylation methodologies reported herein provided a wide library of functionalized imidazo[1,2-*a*]pyridines with a broad susbtrate scope and satisfactory yields. The strategies involved organic or organometallic catatylic systems, but also innovative techniques such as the use of perovskite catalysts and carbon nitride nanosheets, or photocatalyst-free conditions.

#### 2.1.3. Carbonylalkylation and Carbonylation of Imidazopyridines

As a continuation of the visible-light C–H alkylation of imidazopyridines, the addition of carbonyl groups and their derivatives was also widely studied. In 2018, Zhu and Le conducted the visible-light-mediated carbonylalkylation of imidazo[1,2-*a*]pyridines with *N*-arylglycine esters (Figure 20) [55]. The coupling reaction between these two molecules was carried out with a copper catalyst (Cu(OTf)_2_) in acetonitrile. The imidazo[1,2-*a*]pyridine scope investigation indicated that electron-poor substituents increased the transformation efficiency more than the methyl groups. Studying the N-arylglycine esters showed good suitability with a large array of substrates.

The same authors recently extended their synthetic method by coupling imidazo[1,2-*a*]pyridines with α-amino ketones and α-amino acid derivatives [56]. Some improvements were applied: the metal catalyst was replaced by an organophotocatalyst (Eosin Y), with citric acid monohydrate as an additive. Ethanol was employed as a greener solvent, and the visible-light irradiation was monitored with an 18 W blue-LED light (Figure 21). The scope examination of *N*-arylglycine ethyl esters indicated the high reaction efficiency with electron-donating groups on the aryl motif, while various esters (methyl, isopropyl, *tert*-butyl, and benzyl esters) displayed good compatibility with moderate-to-good yields. α-amino ketones delivered the expected imidazo[1,2-*a*]pyridines with low yields. Regarding the scope of imidazo[1,2-*a*]pyridines, a similar trend was observed with a better reactivity of electron-rich substrates. The authors performed control experiments, including radical trapping, reactions with imine substrates, and cyclic voltammetry, to understand the mechanism. The possible path could proceed by an SET between the excited state of eosin Y and the α-amino carbonyl derivative. The formed radical cation could be oxidized into the iminium intermediate, which could undergo an electrophilic addition from the imidazo[1,2-*a*]pyridine. A final oxidation step could provide the desired product.

In 2022, Jiang and Yu realized the ethoxy-carbonyl methylation of imidazo[1,2-*a*]pyridines with α-bromoesters in water, employing rhodamine B (RhB) as the photocatalyst, dilauroyl peroxide as the oxidant, and potassium ethyl xanthogenate as an additive [57]. This method allowed for the preparation of three imidazopyridines with moderate yields. The photochemical reaction was also successfully applied to the preparation of Zolpidem in one step, with 2-bromo-*N*,*N*-dimethylacetamide as the substrate partner (Figure 22).

Another application of the carbonylalkylation reaction was performed by the Chaubey group, with the total synthesis of Zolpidem [58]. After a detailed methodology for the C_3_-carbonylation of imidazo[1,2-*a*]pyridines in the presence of dialkyl malonates, the authors discovered a rapid multi-step synthetic route to Zolpidem in high yields. This sequence was based on the visible-light-promoted C–H carbonylalkylation of the corresponding imidazopyridine, followed by a Krapcho decarboxylation at 160 °C, hydrolysis, and condensation (Figure 23).

An analogous idea came out in Hajra’s group: by changing the carbonylalkylated source (ethyl diazoacetate) and the photocatalyst ([Ru(bpy)_3_]Cl_2_, with bpy: 2,2′-bipyridyl), they accomplished the C_3_-ethoxycarbonyl methylation of imidazo[1,2-*a*]pyridines (Figure 24) [59]. By studying the scope of imidazo[1,2-*a*]pyridines, a good compatibility was noticed with electron-rich substituents. Surprisingly, the reaction did not occur in the presence of electron-withdrawing groups. A slight modification of the optimized conditions was thus applied: by adding 10 mol% of *N*,*N*-dimethyl-*m*-toluidine, a redox-active additive, the C–H carbonylalkylation ran smoothly with satisfactory results (up to 92% yield). The viability of the methodology was confirmed with the gram-scale preparation of ethyl 2-(2-phenylimidazo[1,2-a]pyridin-3-yl)acetate with a 70% yield (Figure 24, Equation (1)) and the late-stage amidation of a C_3_-substituted compound (Figure 24, Equation (2)).

Similarly, Yu, Tan, and Deng expanded Hajra’s methodology to a wide range of diazo derivatives and imidazo[1,2-*a*]pyridine substrates (28 examples) [60]. Subsequently, the reaction showed its applicability with a gram-scale reaction (Figure 25, Equation (1)) and the Zolpidem preparation (Figure 25, Equation (2)). This strategy allowed shorter and more efficient access to synthetic drugs than Chaubey’s approach (*cf*. Figure 23).

In 2019, Guan and He moved one step beyond the concept of imidazo[1,2-*a*]pyridines’ carbonylation [61]. The direct addition of a carbonyl motif on the imidazo[1,2-*a*]pyridine skeleton was conducted under 32 W CFL irradiation with an oxygen balloon and 9-mesityl-10-methylacridinium perchlorate (Acr^+^-Mes). Using a nitrone derivative as a co-substrate, an aryl entity could be included in the carbonyl group. With the optimized conditions in hand, the scope was scrutinized: imidazo[1,2-*a*]pyridines bearing bromo- or chloro-substituents in position C7 exhibited a higher reaction efficiency (65–66% yield) than the C6-substituted ones (50–54% yield). Concerning the nitrone screening, higher yields were noted with the *meta*- and *para*-substitution on the aryl part than the *ortho-*substitution, probably due to steric hindrance. Next, the reaction mechanism was elucidated with control experiments (radical inhibition, ^18^O-labeling reaction, and Stern–Volmer quenching fluorescence) and the X-ray crystal structure of the *N*-hydroxylamine intermediate. The pathway started from the SET between the excited photocatalyst (Acr^+^-Mes*) and the imidazo[1,2-*a*]pyridine. The nitrone could then be introduced in the imidazo[1,2-*a*]pyridine. Two possible paths could then be identified. The first path could involve deprotonation and nitrosobenzene releasing. The resulting radical could react with the radical oxygen species O_2_^•−^ (formed by an SET with the radical photocatalyst) to generate the carbonylated product. The second path could imply an SET from the radical photocatalyst to the radical nitroso, giving the corresponding *N*-hydroxylamine. A second SET could then occur, leading to a radical *N*-hydroxylamine. As the first path, the decomposition of the *N*-hydroxylamine delivered a nitrosobenzene and the corresponding radical, which could be transformed into the target compound (Figure 26).

Carbonylalkylation and carbonylation of imidazopyridines enabled the introduction of amino acid derivatives in the imidazopyridine’s core. The employed approaches consisted in the use of metal catalysts (Cu(OTf)_2_, [Ru(bpy)_3_]Cl_2_) or organophotocatalysts, in apolar (DCM) or polar protic and aprotic solvents (ACN, Dioxane, MeOH, EtOH). Eco-friendly methods demonstrated their efficiency in aqueous media.

#### 2.1.4. Sulfonylmethylation of Imidazopyridines

In the same way, Zhang and Cui exploited an extension of the imidazopyridines’ alkylation for the sulfonylmethylation reaction [62]. By utilizing bromomethyl sulfones with an iridium photocatalyst ([Ir(ppy)_3_]), a broad range of imidazo[1,2-*a*]pyridines could be functionalized efficiently with satisfactory yields (Figure 27). The transformation is also well suited for diversely substituted bromomethyl sulfones. The mechanism investigation by radical trapping revealed that the transformation path could imply radical intermediates. From this observation, the authors suggested an SET from the excited state of the photocatalyst to the bromomethyl sulfone, to deliver a corresponding sulfomethyl radical. The addition of the latter intermediate to the imidazo[1,2-*a*]pyridine’s core provided the corresponding radical, which could be oxidized via an SET with [Ir(ppy)_3_]^+^. The formed cation could be converted into the expected compound by deprotonation.

#### 2.1.5. Formylation of Imidazopyridines

Formyl functional groups constitute a major moiety in *N*-heterocycles, since they could be key building blocks for synthesizing highly complex molecules. In this frame, the visible-light-induced formylation of imidazo[1,2-*a*]pyridines has recently gained interest. The Hajra team developed mild conditions for the regioselective formylation of imidazo[1,2-*a*]pyridines in position C_3_, with rose bengal as the photoredox catalyst, KI as an additive, and TMEDA as the formylating agent [63]. This reaction is suitable with substrates featuring electron-poor, -rich, or halogenated substituents (up to 95% yield). The transformation was entirely inhibited by achieving control experiments with TEMPO or benzoquinone. The same result was noted by replacing O_2_ (from the air) with an argon atmosphere. With all these observations, the authors proposed the following pathway: by excitation of the photocatalyst (RB), a singlet oxygen (^1^O_2_) could be generated, inducing the formation of the iodine radical. This latter intermediate could oxidize the TMEDA as a radical cation. With the superoxide radical anion, the TMEDA-derived radical cation could be turned into an iminium ion. The electrophilic addition with the imidazo[1,2-*a*]pyridine could then occur, followed by a re-aromatization. Iodine could thus oxidize the TMEDA motif, releasing an iminium ion. Consequently, hydrolysis of the iminium ion could afford the desired formylated imidazo[1,2-*a*]pyridine (Figure 28).

#### 2.1.6. Arylation of Imidazopyridines

Recently, Cui and Wu conducted the visible-light C(sp^2^)–H arylation of heterocycles with hypervalent iodine ylides as the arylating agents, eosin Y as the photocatalyst, and potassium carbonate as the base [64]. Among the synthesized heterocyclic scaffolds, five examples of imidazo[1,2-*a*]pyridines were depicted with satisfactory yields (Figure 29).

Sun et al. reported their research on the regioselective azolylation of imidazo[1,2-*a*]pyridines [65]. The installation of the azole nucleus was mediated by 2-bromoazoles under blue-LED irradiation. The photocatalytic process involved Cy_2_NMe (*N*,*N*-dicyclohexylmethylamine) as an organic base and an iridium photocatalyst ([Ir(ppy)_2_(dtbbpy)]PF_6__,_ with dtbbpy: 4,4′-di-tert-butyl-2,2′-dipyridyl). This synthetic approach furnished 29 C_3_-substituted imidazo[1,2-*a*]pyridines with 28 to 79% yield (Figure 30). Electron-poor groups on the imidazopyridine scaffold diminished the heterocycle’s reactivity, whereas electron-rich substituents favored the reaction’s efficiency. In addition, the authors reported good suitability with diversely substituted bromoazoles, i.e., bromothiadiazole, bromothiophene, and bromofuraldehyde. A radical inhibition of the C_3_-azolylation was also conducted with TEMPO: the target molecule was not detected, pointing out the radical character of the transformation. An oxidative quenching of the bromoazole by the excited state photocatalyst could lead to the corresponding heterocyclic radical, which could be added to the imidazopyridine skeleton. Simultaneously, Ir(IV) could reduce the organic base into a radical amine cation. This latter one could then catch hydrogen radicals to release the desired product.

These two examples showed the wide possibility for functionalizing the imidazopyridine scaffold. The introduction of aryl and heteroaryl motifs in good-to-moderate yields was provided by organocatalyst (Eosin Y) or an iridium complex in aprotic polar solvents.

### 2.2. Formation of C–N Bonds

With the presence of heteroarylamines in a plethora of natural products, C–H amination of heterocyclic structures constitutes a long-lasting interest for organic chemists [66]. Efficient, mild, and regioselective methodologies were addressed, especially the eco-friendly C–H functionalization induced by visible light [67,68].

Within this frame, Adimurthy and co-workers published in 2017 the metal-free C_3_ amination of imidazo[1,2-*a*]pyridines [69]. This synthetic strategy allowed for the introduction of aza-heteroarenes (benzotriazole, pyrazole, imidazole, 1*H*-1,2,4-triazole, 1*H*-benzo[*d*]imidazole, and 1*H*-indazole) to the imidazo[1,2-*a*]pyridine platform. Satisfactory yields were obtained, even with halogenated substituents on both reaction substrates (Figure 31).

Similarly, Zhang and Lei introduced an azole motif in imidazo[1,2-*a*]pyridines at position C_3_. In contrast with the previous method, the C–N bond formation additionally required a metal catalyst ([Co(dmgH)(dmgH_2_)]Cl_2,_ with dmg: dimethylglyoximato) [70]. The corresponding C_3_-functionalized imidazo[1,2-*a*]pyridines were generated with good-to-excellent yields. The scope examination with azoles demonstrated good reaction tolerance by employing pyrazoles, imidazoles, or triazoles. A thorough mechanistic study, including the light on/off experiments, radical trapping, cyclic voltammetry measurements, and DFT (density functional theory) calculations, validated the radical reaction path. The excited state of the organophotocatalyst could allow for an SET to the imidazo[1,2-*a*]pyridine. The generated radical cation species could then undergo a nucleophilic attack of the azole substrate, giving the corresponding radical. Simultaneously, a Co(III) catalyst could oxidize the reduced photocatalyst, releasing back the photocatalyst to its fundamental state. The subsequently formed Co(II) could realize an SET to the radical imidazo[1,2-*a*]pyridine. The target aza-heterocycle could be engendered by deprotonation. Co(I) could be converted back into Co(III) by proton capture and dehydrogenation (Figure 32).

The regioselective C–N bond formation could also be extended to incorporate sulfonamide groups on imidazo[1,2-*a*]pyridines. The Sun group outlined the light-mediated C_3_-sulfonamidation reaction with an iridium photocatalyst ([Ir(ppy)_2_(dtbbpy)]PF_6_) and NaClO as the oxidant [71]. The process was very compatible with imidazo[1,2-*a*]pyridines featuring electron-poor or -rich substituents. By contrast, a significant electronic effect could be remarked with the sulfonamides: methyl, methoxy, and *tert*-butyl derived sulfamides enhanced the yields compared to the chlorinated or brominated ones. Control experiments with TEMPO or 1,1-diphenylethene corroborated the radical mechanism. The oxidative quenching of the photocatalyst’s excited state by NaClO could result in an Ir(IV) complex. This organometallic species could be involved in an SET with the sulfamide to deliver a sulfamido radical, which could react with the imidazo[1,2-*a*]pyridine. Oxidation and deprotonation will transform the produced radical into the desired *N*-heterocyle (Figure 33).

In 2020, Braga and his co-workers performed the azo-coupling of imidazo[1,2-*a*]pyridines with aryl diazonium salts under green LED irradiation [72]. By this strategy, 18 functionalized imidazopyridines were prepared with good-to-excellent yields (up to 99%). The reaction’s viability was validated with a gram-scale synthesis of a diazo derivative (Figure 34, Equation (1)) and the reduction of a diazo imidazo[1,2-*a*]pyridine by zinc in acidic conditions (Figure 34, Equation (2)).

More recently, the visible-light-induced C–H amination of imidazo[1,2-*a*]pyridines was exploited in an environmentally friendly manner with micellar catalysis. Li’s approach was based on the use of amphiphilic surfactants in water, which could constitute micelles by hydrophobic interaction [73]. The core of the micelles could be employed as a micro-reactor, where substrates could be activated. This green procedure needs a hydrophilic cationic *N*-aminopyridinium salt as the amine transfer reagent. The “head” of the pyridinium salt (pyridinium nucleus) could interact with the micelle surface, whereas the amine “tail” was localized in the core (*vide infra*). Sodium dodecyl sulfate (SDS) was chosen as the surfactant, yielding better results during the optimization step. With 2,4,5,6-tetrakis(9*H*-carbazol-9-yl) isophthalonitrile (4CzIPN) as the photocatalyst under blue-LED irradiation, a series of C_3_-aminated imidazo[1,2-*a*]pyridines were provided with good-to-excellent yields (up to 92% yield). The reaction path was unraveled by conducting complementary experiments (radical trapping, light-off procedure, process without surfactant, photocatalyst, or N_2_). In the micelle hydrophobic core, an SET from the excited state of 4CzIPN to the pyridinium salt could lead to the amino radical. A radical addition could then occur on the imidazo[1,2-*a*]pyridine. A second SET could furnish the corresponding cation, which could undergo pyridine-mediated deprotonation (Figure 35).

The formation of C–N bonds in the imidazopyridine’s structure allowed the incorporation of aza-heterocyclic nuclei, sulfonamides, amines, and diazo groups on the C_3_ position. The implied reactions needed organophotocatalysts (Acr^+^-Mes, eosin Y-Na_2_, 4CzIPN) or metal complexes (Co- or Ir-derived catalysts). In a more sustainable way, a micellar system was employed instead of conventional organic solvents. In all the examples, the desired products were obtained with excellent yields.

### 2.3. Formation of C–O Bonds

With the major occurrence of the C–O bond in natural or biologically active compounds, the construction of this motif is highly sought by researchers. Among the developed strategies, Hajra and co-workers investigated a metal-free methodology for the C–H alkoxylation of imidazo[1,2-*a*]pyridines [74]. With an organophotocatalyst (rose bengal) and alcohol under visible-light LED irradiation, the group constructed a C–O bond on the position C_3_ of the imidazo[1,2-*a*]pyridine’s nucleus. Twenty-seven examples of functionalized imidazo[1,2-*a*]pyridines were synthesized, bearing various alcohols. Good-to-excellent yields were obtained with *N*-heterocycles displaying electron-poor or -rich substituents without any electronic effect. In the dark or with a radical inhibitor, control experiments could gain insight into the reaction mechanism: by an SET with rose bengal. An imidazopyridine radical cation could be engendered. This latter intermediate could react with alcohol to yield the corresponding radical. The desired alkoxylated product could then be formed by HO_2_^•^ hydrogen abstraction (Figure 36).

More recently, Singh and his colleagues developed C–H activation mechanism assisted by directing groups for the oxygenation of heterocyclic scaffolds [75]. This original transformation required 1,2,3,5-tetrakis(carbazol-9-yl)-4,6-dicyanobenzene (4CzIPN) as the organic photocatalyst, palladium acetate as the metal catalyst, and potassium trifluoroacetate as the base, in a solvent mixture (CF_3_CO_2_H/DMF: dimethylformamide) and under an oxygen atmosphere. It should be highlighted that a high temperature is needed to oxidize Pd(II) to Pd(IV). In this context, the authors reported one example of imidazopyridine C–H activation, using the imidazopyridine itself as the directing group. In contrast with the previous approach, the imidazopyridine platform is herein functionalized in the aryl part with a 66% yield (Figure 37).

These two examples showed the large possibility of C–O functionalization of imidazopyridines. By varying the conditions, the oxygenated motif could be introduced regiospecifically in different positions on the imidazopyridine’s structure. In metal-free conditions, C_3_-functionalization was realized. In contrast, the palladium-catalyzed reaction allowed for the C–H activation on the aryl part of the scaffold.

### 2.4. Formation of C–P Bonds

The functionalization of imidazo[1,2-*a*]pyridines with phosphorous motifs was also envisioned. In 2020, Sun, Chen, and Yu studied the C_3_-phosphorylation of imidazo[1,2-*a*]pyridines, under visible-light irradiation [76]. By using RhB as a photoredox catalyst, lauroyl peroxide (LPO) as an oxidant, and diethyl carbonate as the reaction solvent, imidazo[1,2-*a*]pyridines were efficiently phosphorylated in position C_3_ with good yields. The transformation presented a good tolerance in the presence of electron-withdrawing or -donating groups on the imidazo[1,2-*a*]pyridine scaffold. With respect to the phosphine oxides, variously substituted diaryl phosphine oxides could be employed in the reaction conditions. After the scope investigation, the authors decided to deepen their knowledge of the C_3_-phosphorylation by examining its pathway. Mechanistic insights (radical trapping, Stern–Volmer voltammetry fluorescence quenching, and variation of standard conditions) suggested an energy transfer (ET) from RhB* to the imidazo[1,2-*a*]pyridine substrate. The substrate could then evolve to a triplet state (T), capable of reacting with LPO for generating a phosphine radical through a radical cascade. The addition of phosphine on the imidazo[1,2-*a*]pyridine structure’s radical could thus occur. Final deprotonation could deliver the expected phosphorylated aza-heterocycle (Figure 38).

### 2.5. Formation of C–S Bonds

As sulfur-containing molecular architectures are widely available in various drugs, biologically active molecules, or natural compounds [77], the conception of C–S bonds has received growing attention from the chemistry community. Complementary to the conventional metal-catalyzed cross-coupling methods [78,79,80], researchers have sought milder procedures for C–S bond creation [81]. Following the examples mentioned above in C–C and C–heteroatom bond formation, Yang and Wang envisaged the C_3_-sulfenylation of imidazo[1,2-*a*]pyridines under visible light irradiation [82]. By utilizing eosin B as the organophotocatalyst, *tert*-butyl hydroperoxide (TBHP) as the oxidant, and aryl sulfinic acids as the sulfur source, functionalized imidazo[1,2-*a*]pyridines were delivered smoothly (64 to 87% yield). No electronic influence was remarked for either electron-withdrawing or -donating substituted sulfinic acids. The deciphering of the reaction path by control experiments brought to light the radical character of the transformation. The photoexcited species eosin B* could realize an SET with *t*BuOOH. Radical *t*BuOO^•^ could then abstract a hydrogen atom to the sulfinic aryl acid, evolving into a thiyl radical by reduction. This intermediate could thus be added to the imidazo[1,2-*a*]pyridine. Finally, the target product could be obtained by an SET and deprotonation (Figure 39).

In 2018, Barman and co-workers proposed an analogous synthetic methodology for the C_3_-sulfenylation of imidazo[1,2-*a*]pyridines [83]. Compared to the preceding example, thiols replaced sulfinic acids as the sulfenylating agent. The easier procedure was reported, since the oxidation step was conducted by ambient air. A good reaction efficiency was noticed for many imidazo[1,2-*a*]pyridines and thiols exhibiting electron-poor or -rich groups (Figure 40). The method’s viability was checked in a gram-scale transformation with 2-phenylimidazo[1,2-*a*]pyridine and thiophenol, affording the expected compound an 87% yield.

Motivated by the C–H thiocyanation promoted by Hajra et al. [33], Tang and Yu combined photochemistry with heterogeneous catalysts. As the microporous polymer catalysts could be efficiently recycled in reactions, the authors employed the benzo[1,2-*b*:4,5-*b’*]dithiophene-4,8-dione conjugated microporous polymer (CMP-BDD) as a heterogeneous photocatalyst. With this strategy, the C–H thiocyanation of imidazo[1,2-*a*]pyridines has been revived in a greener manner [84]. Similar to the Hajra group’s results, a good tolerance was noticed for imidazopyridines featuring electron-donating or withdrawing substituents (Figure 41).

Following the same approach, Chen and Yu carried out the C(sp^2^)–H thiocyanation of heterocyclic compounds with carbon nitride (g-C_3_N_4_) as the heterogeneous photocatalyst [85]. Under blue-LED irradiation and with a green solvent (dimethyl carbonate), thiocyanated imidazo[1,2-*a*]pyridines were provided in good yields (up to 96% yields). As previously mentioned, the transformation was very compatible, with substrates displaying methyl, methoxy, fluorine, or thienyl groups (Figure 42).

Oxidative sulfur forms could also be incorporated into the imidazo[1,2-*a*]pyridine’s structure. Piguel et al. accomplished the light-induced regioselective sulfonylation of imidazopyridines in the presence of DABCO-*bis*(sulfur dioxide) and an aryl iodonium salt [86]. Aside from forming the C–S bond, this reaction allowed the integration of an aryl group on the sulfone part. The scope examination with respect to the imidazo[1,2-*a*]pyridine substrates indicated neglectable electronic effects of the *N*-heterocycle substituents, considering the good yields obtained. The same trend was found by varying aryl iodonium hexafluorophosphates. A radical path was suggested by mechanistic insights (Stern–Volmer fluorescence quenching experiments, light-off reactions, and radical trapping). The green LED activation could favor the formation of excited species of Eosin Y*. An SET could occur with the aryl iodonium salt, producing the aryl radical, which could be trapped by the DABCO-*bis*(sulfur dioxide). The transfer of sulfonyl radicals on the imidazo[1,2-*a*]pyridine could thus follow. The second SET and deprotonation could afford the final sulfonylated *N*-heterocycle (Figure 43).

The formation of C–S bonds in the imidazopyridine skeleton permitted the synthesis of highly functionalized imidazopyridines. By these presented methods, thioethers, thiocyanates, and sulfonyl derivatives could be provided in good yields. In addition to the sulfur sources (e.g., sulfinic acids, thiols), these reactions involved organophotocatalysts (eosin B, eosin Y, rose bengal). Some transformations were optimized by using a greener solvent (dimethyl carbonate) combined with recyclable catalysts (microporous polymer or carbon nitride).

### 2.6. Formation of C–Se Bonds

Selenylated molecules have attracted great interest in chemistry, namely by their biological and medicinal properties including anticancer or anti-Alzheimer’s activities. They also represent key synthetic intermediates/substrates in total synthesis or asymmetric catalysis [87]. The synthesis of organoselenium compounds has been an important topic in organic chemistry with their rising importance. In the pursuit of eco-compatible processes, Liu et al. described in 2017 the first visible-light-promoted C_3_-selenylation under aerobic conditions [88]. Three selenylated imidazo[1,2-*a*]pyridines were elaborated, in the presence of diphenyl diselenide and FIrPic (bis[2-(4,6-difluorophenyl)pyridinato-C2,*N*](picolinato) iridium(III)). Because of their low electron density, the imidazo[1,2-*a*]pyridines were only afforded in moderate yields. Mechanistic investigations with TEMPO and photoluminescence experiments pointed out the SET from the excited state of FIrPic to the diphenyl diselenide, leading to the PhSe^•^ radical. PhSe^•^ was then oxidized into PhSe^+^, which could undergo an electrophilic addition to the imidazo[1,2-*a*]pyridine’s structure. The expected molecule could be furnished by deprotonation (Figure 44).

In 2018, Braga et al. optimized Liu’s protocol by replacing FIrPic with an organophotocatalyst (rose bengal) [89]. The group slightly improved the reaction yield with the imidazo[1,2-*a*]pyridine scaffold (Figure 45).

In the same trend, the Kumaraswamy team envisaged the C_3_-selenylation of an imidazo[1,2-*a*]pyridine motif with diphenyl selenide as the selenylating reagent, LiCl as an additive, and 2-methyl-1-propanol as a solvent [90]. Photoactivation of diphenyl selenide provided the desired N-heterocycle with a 55% yield (Figure 46).

In 2019, Yasuike and co-workers conceptualized an alternative light-induced method with ammonium iodide instead of FIrPic as the photocatalyst [91]. A series of 3-(arylselanyl)imidazopyridines were prepared with good-to-excellent yields under aerobic conditions (Figure 47). The reaction was very compatible with variously substituted diarylselenides.

The incorporation of selenylated motifs was accomplished with various visible-light-induced methodologies. A narrow scope of susbtrates was explored due to the low diversity of the selenylated source. An iridium-based photocatalyst or rose bengal could be employed for these reactions. As an alternative metal-free approach, halogenated salts (LiCl or NH_4_I) were used for the selenation of imidazopyridines, with moderate-to-good yields.

### 2.7. Formation of C–Br Bonds

In 2019, Lee, Jung, and Kim reported the C_3_-bromination of imidazo[1,2-*a*]pyridines under visible light [92]. The transformation required CBr_4_ as a bench-stable bromine source and an iridium-derived photocatalyst (Figure 48). The scope examination showed that imidazo[1,2-*a*]pyridines presenting electron-withdrawing or -donating groups allowed for the preparation of brominated aza-heterocycles with satisfactory yields. The process’s viability was validated with a gram-scale reaction, providing the expected functionalized imidazo[1,2-*a*]pyridine with an 83% yield.

The C–H functionalization of imidazopyridines constitutes a powerful approach for the synthesis of highly substituted imidazopyridines. All the strategies proved their efficiency with broad scopes, wide functional group tolerances, and high yields. The viability of these methods was validated with gram-scale reactions and the preparation of compounds with biological interest. These methodologies mainly used organophotocatalysts, ruthenium, or iridium complexes under blue-LED irradiation. Some of these transformations have been improved in an eco-compatible way by employing green solvents or sustainable materials as heterogeneous catalysts.

Several methodologies have emerged in parallel with the C–H functionalization strategies to elaborate these scaffolds, especially the multi-component one-pot reactions.

## 3. Multi-Component Reactions

In 2018, Siddiqui et al. reported a “green” Groebke–Blackburn–Bienaymé reaction to prepare imidazo[1,2-*a*]pyridines, in the presence of 2-amino-pyridines, aldehydes, and isocyanides. The main benefit of this strategy relies on the absence of solvents and metalz. The transformation only requires the use of a CFL delivering visible light, affording the corresponding *N*-heterocycles (11 examples) with high yields. A good tolerance was observed for electron-poor and -rich substituents. The pathway suggested the formation of an imine intermediate, which would undergo a nucleophilic addition by the isocyanide. Through an intramolecular cyclization, the generated amine would be photoactivated (Figure 49) [93].

More recently, the Singh team established a similar procedure for constructing imidazo[1,2-*a*]pyridines, with 2-amino-pyridines, benzyl amines, and *tert*-butyl isocyanide. In contrast with the previous strategy, eosin Y was employed as a photocatalyst, and a mixture of eco-friendly solvents (EtOH and water) was needed for this multi-component reaction. By irradiation under visible light at room temperature with 22 W white LEDs, a wide range of substituted imidazo[1,2-*a*]pyridines were obtained with excellent yields (Figure 50). The cyclization was very compatible with benzylamine, displaying electron-withdrawing or -donating groups. However, the scope exploration with respect to aminopyridines was only limited to 2-aminopyridine and 5-bromo-2-aminopyridine. Mechanistic insights confirmed the radical pathway and the importance of oxygen in the open flask transformation. Based on these observations, the reaction path would proceed through a hydrogen atom transfer (HAT) induced by the photoexcitation of eosin Y. A hydrogen atom would be extracted from the benzylamine, producing the corresponding benzylic amine. This latter intermediate would be oxidized with O_2_ into benzylimine. The nucleophilic addition of aminopyridine to benzylimine would form the aminal and subsequently the imine. As before, the imine undergoes a nucleophilic attack of the isocyanide and a light-promoted cyclization to deliver the expected imidazo[1,2-*a*]pyridine [94].

In a solvent-free medium, the same group pursued their research with a greener methodology and with styrene derivatives and 2-aminopyridines [95]. By adding *tert*-butyl isocyanide and eosin Y under blue-LED irradiation, they obtained a series of diversely substituted imidazo[1,2-*a*]pyridines with 92 to 95% yield (Figure 51). The transformation was well suited for electron-poor and -rich groups.

Das and Thomas developed a one-pot synthesis of imidazo[1,2-*a*]pyridines involving alkenes, *N*-bromosuccinimide (NBS), and 2-aminopyridine [96]. The substrate scope only included the variation of the styrene substrates. Higher yields were noticed with electron-rich substituents (R = Me or OMe) compared to the electron-withdrawing ones (Br, Cl, NO_2_, CO_2_H, and CO_2_Et). The viability of the process was validated with a gram-scale reaction starting from styrene (R=H), furnishing the desired compound with a 74% yield. A pathway examination established the influence of the photoactivation on the reaction and the bromoketone formation as a key intermediate. The suggested mechanism would consist of a double addition of a bromo radical, provided by NBS under photoactivation, on the styrene. The formed bromoketone would then be subjected to a nucleophilic attack by the 2-aminopyridine. By light induction, the carbonyl moiety would serve as a photosensitizer, allowing the production of a diradical intermediate. After the radical cyclization, the abstraction of water and the tautomerization of the imidazopyridine would afford the suitable isomer (Figure 52).

In brief, multicomponent reactions demonstrated their applicability in eco-compatible conditions. By using organocatalyst and mainly solvent-free conditions, the preparation of diversely substituted imidazopyridines was achieved with good yields (up to 95% yield).

Complementary to the multi-component transformations, various methods have been established to synthesize the imidazo[1,2-*a*]pyridines.

## 4. Miscellaneous Reactions

In 2016, Singh and co-workers reported an oxidative photoredox catalysis to conceive 2-nitro-3-arylimidazo[1,2-*a*]pyridines in a regioselective manner. To that end, nitrostyrenes and 2-aminopyridines were used as substrates and eosin Y as a photoredox catalyst (Figure 53). The irradiation with a green LED at room temperature, under an open atmosphere, and in acetonitrile led to the expected 2-nitro-3-arylimidazo[1,2-*a*] pyridines with good yields (67 to 78% yield). It should be highlighted that nitrostyrenes exhibiting electron-withdrawing or -donating substituents afforded imidazo[1,2-*a*]pyridines in satisfactory yields. In contrast, the desired compounds were not obtained with aliphatic nitrostyrenes. The scope for the 2-aminopyridines was only limited to the methyl-substituted substrates. However, the substituent position was studied, demonstrating an insignificant influence of the methyl group on the reaction yield. Mechanistic investigations indicated that the transformation should involve a Michael addition between the 2-aminopyridine and the nitrostyrene, followed by an SET induced by eosin Y under visible light, an intramolecular cyclization, and an oxidation step [97].

Kamal et al. described the visible-light-induced coupling of α-keto vinyl azides and 2-aminopyridines, with [Ru(bpy)_3_]Cl_2_·6H_2_O as a photocatalyst [98]. The scope concerning α-keto vinyl azides provided diversely substituted imidazo[1,2-*a*]pyridines with excellent yields. 2-aminopyridines bearing chloro, methyl ester, or methyl substituents also led to good yields for the expected *N*-heterocycles. Complementary studies hinted that the pathway would imply a photo-decomposition of the vinyl azides into an azirine intermediate. Moreover, the prepared imidazo[1,2-*a*]pyridines were biologically evaluated on different cancer cell lines: A549 (lung cancer), DU-145 (prostate cancer), MCF-7 (breast cancer), and Hela (cervical cancer)). Some of these molecules presented encouraging cytotoxic activities against A549, DU-145, and MCF-7 cell lines (Figure 54).

The Chuah group reported a visible-light-induced cyclization to form the imidazo[1,2-*a*]pyridine motif. The reaction necessitated β-ketoesters and 2-aminopyridines as substrates, erythrosin B as a photoredox catalyst, and KBr as a halogenating agent. A wide range of β-ketoesters was well tolerated, affording the corresponding products in good yields. The variation of the 2-aminopyridines allowed access to imidazo[1,2-*a*]pyridines with a 59 to 85% yield. Mechanism understanding was carried out with control experiments and cyclic voltammetry. The results indicated that the light irradiation would promote an SET from the bromide ion (Br^−^) to the excited photocatalyst, generating a bromine radical. Alongside the photoredox catalysis, a condensation reaction would occur between the β-ketoester and the 2-aminopyridine to obtain the enamine intermediate. This latter compound would react with the bromine radical. By proton abstraction with O_2_^•−^, an α-bromo ketone would then be generated, which would undergo an intramolecular cyclization (Figure 55) [99].

Recently, the Sun team depicted the photoredox synthesis of C_3_-alkylated imidazo[1,2-*a*]pyridines with α-bromocarbonyls and 2-aminopyridines [100]. The methodology consisted of a one-pot condensation and alkylation with an iridium-based photocatalyst under blue-light activation. Satisfactory yields were obtained with a wide range of α-bromocarbonyls (Figure 56, Equation (1)). The extension of the strategy was explored with multi-component reactions involving α-bromocarbonyls, alkyl bromides, and 2-aminopyridines substrates. Again, the elaboration of “unsymmetrical” C_3_-alkylated imidazo[1,2-*a*]pyridines was achieved with good yields (Figure 56, Equation (2)). The whole approach was validated with a gram-scale reaction and the preparation of Zolpidem, a drug used for insomnia treatment. In light of control experiments, the reaction path would proceed via an alkylation–condensation sequence, followed by the addition of a radical-derived acetophenone, generated by an SET, on the imidazo[1,2-*a*]pyridine’s nucleus.

Imidazo[1,5-*a*]pyridine derivatives could also be built through the photocyclization of imidazoles at room temperature and in NMP [101]. This approach was based on the light irradiation of imidazoles, displaying strong electron-rich substituents in position 2 (Figure 57). A total of five imidazo[1,5-a]pyridine-5,8-diones were synthesized with moderate yields, probably due to photodegradation. The reaction path suggested a ring contraction of the pyrone ring followed by a cyclopentanedione ring opening. The resulted biradical intermediate would undergo decarbonylation, intramolecular cyclization, and oxidation, yielding the expected heterocycle.

All these syntheses illustrated the broad range of the possible methods for constructing the imidazopyridine’s structure. Satisfactory yields were obtained for the prepared highly substituted imidazopyridines. The applicability of some strategies was confirmed with the elaboration of imidazo[1,2-*a*]pyridines with biological interest.

## 5. Conclusions

In summary, all the presented studies for the construction of imidazopyridines have reflected the strong emergence of photochemistry over the last decade. Under visible-light photocatalysis, efficient access to scaffolds with a high biological interest is achievable, with good yields and a wide functional group compatibility. These methodologies also represent promising alternatives to classical approaches for elaborating these motifs, with their mild and eco-friendly conditions (such as room temperature or green solvents). The transformations involved in the synthesis of these *N*-heterocycles could be grouped into three categories: (1) the C–H functionalizations, which mainly occur in the C_3_-position of the imidazo[1,2-*a*]pyridines, for the formation of carbon–carbon or carbon–heteroatom bonds, (2) the multi-component reactions, essentially based on the Groebke–Blackburn–Bienaymé reaction, and (3) the cyclization reactions, allowing for the preparation of highly substituted imidazo[1,2-*a*]pyridines and imidazo[1,5-*a*]pyridines. By complementary studies, especially radical trapping, the reactions pathways shed light on the radical mechanisms.

Therefore, this review constitutes a valuable tool for synthetic chemists working in this exciting field. We also hope that this overview will inspire novel synthetic strategies using visible-light activation, paving the way for constructing a similarly original kind of heterocyclic moieties. In parallel, the development of more environmentally benign procedures should be pursued in the future, namely with the employment of reusable catalysts and easy-handling substrates or a reduction in organic waste.

## Data Availability

Not applicable.

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
