# Peer review of "Recent Developments in the Photochemical Synthesis of Functionalized Imidazopyridines"

_molecules, 2022, doi:10.3390/molecules27113461_

Round 1

Reviewer 1 Report

Imidazopyridines are important scaffolds in organic chemistry and many methods have been developed for their synthesis. Among them photochemical methods are also contributed significantly. Photochemistry offers many advantages over conventional heating, sometimes it delivers the functionalized molecules which couldn’t be possible with conventional heating. In this context, the current review “visible-light induced approaches for synthesizing and functionalizing organic molecules” is timely and important. The examples are collected rather complete, the summary is logical and clear. To my knowledge, there is no other review with specifically describes photochemical synthesis of imidazopyridines content in recent years. From these, I am pleased to recommend the acceptance of this review after minor revisions.

  • In Line number 29 after the “to prepare these N-heterocycles”. Recent reviews on synthesis of Imidazo1,2a] pyridines must be cited.
  • Figure 1 specifically shows structures of Imidazo1,2a] pyridines scaffolds so, in the title imidazopyridines should read Imidazo1,2a] pyridines.
  • Line 48 “imidazopyridines building” should read imidazopyridine building blocks.
  • Change the numbers to subscript in lines 66,72,83,86,87,92…… check throughout the article.
  • Line 174, “molecule” should read “intermediate”.
  • Line 180, “abovementioned” should read “above mentioned”
  • Scheme 14 should be moved to arylation reactions.
  • Line 423, “tcould” should read could.
  • Line 482, zing should read “Zinc”
  • Line 521, assistedby should read assisted by

Author Response

Thank you very much for all the comments. Please see below our answers for your requests:

Point 1: In Line number 29 after the “to prepare these N-heterocycles”. Recent reviews on synthesis of Imidazo1,2a] pyridines must be cited.

Response 1: We added the following reference (Ref. 9) in the paper: Chavana, K.H.; Kedar, N.A. Recent Developments in Synthesis of Imidazo[1,2-a] pyridines (2016-2020). Chemistry & Biology Interface 2021, 11, 34-39. This reference is added as a complement to the recent reviews on the synthesis of Imidazo[1,2a]pyridine throughout the paper (Ref 1, 14, 17).

Point 2: Figure 1 specifically shows structures of Imidazo1,2a] pyridines scaffolds so, in the title imidazopyridines should read Imidazo1,2a] pyridines.

Response 2: We change the figure name to "Examples of drugs and biologically relevant compounds displaying an imidazo[1,2-a]pyridine scaffold".

Point 3: Line 48 “imidazopyridines building” should read imidazopyridine building blocks.

Response 3: We corrected according to the remark "the imidazopyridines building blocks".

Point 4: Change the numbers to subscript in lines 66,72,83,86,87,92…… check throughout the article.

Response 4: The numbers were changed to subscript in the following lines: 66, 72, 83, 86, 87, 92, 97, 98, 111, 113, 114, 118, 119, 121, 130, 131, 136, 139, 144, 150, 152, 156, 182, 183, 199, 204, 341, 351, 359, 413, 427, 440, 446, 448, 449, 463, 466, 477, 485, 496, 504, 511, 520, 534, 537, 542, 550, 557, 569, 571, 579, 613, 620, 631, 636, 637, 642, 649, 651, 659, 763, 769, 776, 891, 888, 908, 933, 954, 1021. 

Point 5: Line 174, “molecule” should read “intermediate”.

Response 5: The word "molecule" was replaced by "intermediate" as suggested.

Point 6: Line 180, “abovementioned” should read “above mentioned”

Response 6: The word was corrected as such in line 180.

Point 7: Scheme 14 should be moved to arylation reactions.

Response 7: Scheme 14 was moved to page 25 in the " Arylation of imidazopyridines" part.

Point 8: Line 423, “tcould” should read could.

Response 8: The word was corrected as such in the manuscript.

Point 9: Line 482, zing should read “Zinc”

Response 9: The word was corrected as such in the manuscript.

Point 10: Line 521, assistedby should read assisted by

Response 10: The word was corrected as such in the manuscript.

Reviewer 2 Report

The review “Recent developments in the photochemical synthesis of functionalized imidazopyridines” describes the recently published papers concerning the synthesis of highly functionalized imidazopyridines by light induction.

The paper presents important aspects concerning an important scaffold for both organic and medicinal chemistry. This represent the backbone of several drugs with hypnotic, therapy for insomnia and gastroprotective effect. Moreover, several activities like antibacterial, analgesic, anti-inflammatory, antiviral, and antitumor ones were evidenced for this condensed moiety.

The efficient and eco-friendly processes used for highly functionalised imidazopyridines synthesis such as photocatalytic methods with metallic or metal-free systems are clear and detailed described. A broad range of methods like C-H functionalization, condensation or multi-component reactions were systematic presented.

I consider that all these aspects are important considering the biological importance of imidazopyridines and the possibility to develop strategies eco-friendly that could provide highly functionalised scaffold in good yield.

I therefore recommend minor revision having in view the following aspects:

- The references in text must be provided before the point and with break (i.e. [5]. instead of.[5])

- The corresponding references must be added at the Schemes together with the working team.

- All abbreviations must be detailed (i.e. DMSO, TEMPO, TMEDA, BHT, DABCO, TfOH, DMF) and the ligands from complexes used as catalysts (bpy, ppy, dmgH) in text.

- The expressions “(reaction under nitrogen atmosphere or in open-air flask, radical trapping with TEMPO…)” at row 260 and “(anticancer or anti-Alzheimer activities, antioxidant, apoptosis induction…)” at row 616 are incomplete.

In Scheme 25 the structure of Ru(bpy)3Cl2 must be corrected (there is an additional link between two bpy) and the complex formula must be corrected as  [Ru(bpy)3]Cl2 both in text and Scheme. Otherwise all complexes must be provided in correct form with evidence the ligands in square brackets as exemplified above.

  • The complex [Ir(ppy)3] at row 397 not contain any anion having in view that at row 404 is presented as [Ir(ppy)3]+?
  • The Ir photocatalyst used by Sun group [70] must be provided.
  • The zing must be replaced by zinc I suppose (row 482).
  • The rhodamine B was abbreviated as RhB at row 535 but appears in text before. As result the abbreviation must be provided first when appear in text and then this must be used.
  • The complex (bis[2-(4,6-difluoro-phenyl)pyridinato-C2,N](picolinato) iridium(III) abbreviated FIrPic (row 622) cannot contain Ir(III) but Ir(I) if this not contain other two anions for neutrality (see also Scheme 44).

A comparative discussion must be added at each subsection.

Author Response

Thank you very much for all the comments. Please see below our answers for your suggestions:

Point 1: The references in text must be provided before the point and with break (i.e. [5]. instead of.[5])

Response 1:  All the references in text were corrected according to the comments.

Point 2: The corresponding references must be added at the Schemes together with the working team.

Response 2:  The corresponding references were added to the schemes.

Point 3: All abbreviations must be detailed (i.e. DMSO, TEMPO, TMEDA, BHT, DABCO, TfOH, DMF) and the ligands from complexes used as catalysts (bpy, ppy, dmgH) in text.

Response 3: All the abbreviations (solvents and ligands) were detailed throughout the manuscript as suggested.

Point 4: The expressions “(reaction under nitrogen atmosphere or in open-air flask, radical trapping with TEMPO…)” at row 260 and “(anticancer or anti-Alzheimer activities, antioxidant, apoptosis induction…)” at row 616 are incomplete.

Response 4:  The expression “reaction under nitrogen atmosphere or in open-air flask, radical trapping with TEMPO” was replaced by “namely reaction under nitrogen atmosphere or in open-air flask, radical trapping with TEMPO” and the sentence “Selenylated molecules have attracted great interest in chemistry, namely by their biological and medicinal properties (anticancer or anti-Alzheimer activities, antioxidant, apoptosis induction…).” was replaced by “Selenylated molecules have attracted great interest in chemistry, namely by their biological and medicinal properties including anticancer or anti-Alzheimer activities.”

Point 5: In Scheme 25 the structure of Ru(bpy)3Cl2 must be corrected (there is an additional link between two bpy) and the complex formula must be corrected as  [Ru(bpy)3]Cl2 both in text and Scheme. Otherwise all complexes must be provided in correct form with evidence the ligands in square brackets as exemplified above.

Response 5:  The structure and the formula (text and scheme) of [Ru(bpy)3]Cl2 were corrected in the manuscript. All the other complexes were corrected in the text and the schemes with ligands in square brackets.

Point 6: The complex [Ir(ppy)3] at row 397 not contain any anion having in view that at row 404 is presented as [Ir(ppy)3]+?

Response 6: The complex [Ir(ppy)3] did not contain any anion. In contrast, the generated complex [Ir(ppy)3]+ probably has a counter anion Br-, coming from the bromomethyl sulfone substrate.

Point 7: The Ir photocatalyst used by Sun group [70] must be provided.

Response 7: The Ir photocatalyst ([Ir(ppy)2(dtbbpy)]PF6) was provided previously by the same group for the regioselective azolylation of imidazo[1,2-a]pyridines.

Point 8: The zing must be replaced by zinc I suppose (row 482).

Response 8: The word was corrected as such in the manuscript.

Point 9: The rhodamine B was abbreviated as RhB at row 535 but appears in text before. As result the abbreviation must be provided first when appear in text and then this must be used.

Response 9: The abbreviation was added firstly as such and used after in the manuscript.

Point 10: The complex (bis[2-(4,6-difluoro-phenyl)pyridinato-C2,N](picolinato) iridium(III) abbreviated FIrPic (row 622) cannot contain Ir(III) but Ir(I) if this not contain other two anions for neutrality (see also Scheme 44).

Response 10: After double-checking the original paper of Liu and co-workers, the complex exactly mentioned in their research article and supporting information was “FIrPic (bis[2-(4,6-difluorophenyl) pyridinato-C2,N](picolinato) iridium(III))”.

Point 11: A comparative discussion must be added at each subsection.

Response 11: For the fluoroalkylation of imidazopyridines part, this paragraph was added: “In summary, fluoroalkylation of imidazopyridines could be reached in several reaction conditions, in moderate to good yields (up to 96% yield). All the approaches described here required polar aprotic solvents (mainly ACN, DMSO) and organic bases or acids. The photocatalysts employed were organophotocatalysts or fac-[Ir(ppy)3]. These strategies allowed access to diversely fluorinated compounds.”

For the alkylation of imidazopyridines part, this paragraph was added: “Summarily, the alkylation methodologies reported herein provided a wide library of functionalized imidazo[1,2-a]pyridines with a broad susbtrate scope and satisfactory yields.  The strategies involved organic or organometallic catatylic systems, but also innovative techniques such as the use of perovskite catalyst and carbon nitride nanosheet, or photocatalyst-free conditions.”

For the carbonylalkylation and carbonylation of imidazopyridines part, the following paragraph was added: “Carbonylalkylation and carbonylation of imidazopyridines enabled the introduction of amino acids derivatives in the imidazopyridine core. The employed approaches consisted in the use of metal catalysts (Cu(OTf)2 , [Ru(bpy)3]Cl2) or organophotocatalysts,  in apolar (DCM) or polar protic and aprotic solvents (ACN, Dioxane, MeOH, EtOH). Eco-friendly methods demonstrated their efficiency in aqueous media.”

For the arylation of imidazopyridines part, the following paragraph was added: “These two examples showed the wide possibility for functionalizing the imidazopyridine scaffold. The introduction of aryl and heteroaryl motifs in good to moderate yields was provided by organocatalyst (Eosin Y) or an iridium complex, in aprotic polar solvents.”

For the formation of C-N bonds part, the following sentences were added: “The formation of C-N bonds in the imidazopyridine structure allowed the incorporation of aza-heterocyclic nuclei, sulfonamides, amines and diazo groups on the C3 position. The implied reactions needed organophotocatalysts (Acr+-Mes, eosin Y.Na2, 4CzIPN) or metal complexes (Co or Ir-derived catalysts. In a more sustainable way, micellar system was employed instead of conventional organic solvents. In all the examples, the desired products were obtained in excellent yields.”

For the formation of C-O bonds part, we added the following paragraph: “These two examples showed the large possibility of C-O functionalization of imidazopyridines. By varying the conditions, the oxygenated motif could be introduced regiospecifically in different positions on the imidazopyridine structure. In metal-free conditions, the C3-functionalization was realized. In contrast, the palladium-catalyzed reaction allowed the C-H activation on the aryl part of the scaffold.”

For the formation of C-S bonds part, this paragraph was added: “The formation of C-S bonds in the imidazopyridine skeleton permitted the synthesis of highly functionalized imidazopyridines. By these presented methods, thioethers, thiocyanates, sulfonyl derivatives could be provided with good yields. In addition to the sulfur sources (e. g. sulfinic acids, thiols), these reactions involved organophotocatalysts (eosin B, eosin Y, rose bengal). Some transformations were optimized by using greener solvent (dimethyl carbonate) combined with recyclable catalysts (microporous polymer or carbon nitride).”

For the formation of C-Se bonds part, this paragraph was added: “The incorporation of selenylated motifs was accomplished with various visible-light induced methodologies. A narrow scope of substrates was explored, due to the low diversity of the selenylated source. Iridium-based photocatalyst or rose bengal could be employed for these reactions. As an alternative metal-free approach, halogenated salts (LiCl or NH4I) were used for the selenylation of imidazopyridines, in moderate to good yields.”

For the C-H functionalization part, the following paragraph was added: “The C-H functionalization of imidazopyridines constitutes a powerful approach for the synthesis of highly substituted imidazopyridines. All the strategies proved their efficiency with broad scopes, wide functional group tolerance and high yields. The viability of these methods was validated with gram-scale reactions and the preparation of compounds with biological interest. These methodologies mainly used organophotocatalysts, ruthenium or iridium complex under blue LED irradiation. Some of these transformations have been improved in an eco-compatible way by employing green solvents or sustainable materials as heterogeneous catalysts.”

For the multicomponent reactions part, these sentences were added: “In brief, multicomponents reactions demonstrated their applicability in eco-compatible conditions. By using organocatalyst and mainly solvent-free conditions, the preparation of diversely substituted imidazopyridines was achieved with good yields (up to 95% yield).”

For the miscellaneous reactions part, we added this paragraph: “All these syntheses illustrated the broad range of the possible methods for constructing the imidazopyridine structure. Satisfactory yields were obtained for the prepared highly substituted imidazopyridines. The applicability of some strategies was confirmed with the elaboration of imidazo[1,2-a]pyridines with biological interest.